# A Critical Review of Extraction Methods for Vanadium from Petcoke Ash

**Hari Jammulamadaka and Sarma V. Pisupati** *

Center for Critical Minerals, John and Willie Leone Family Department of Energy and Mineral Engineering, EMS Energy Institute, 407 Academic Activities Building, The Pennsylvania State University, University Park, PA 16802, USA
* Correspondence: sxp17@psu.edu

**Abstract:** Petcoke is a solid carbon-rich residue produced during petroleum refining. Petcoke mineral matter is rich in vanadium that, when alloyed with other metals, can significantly improve its properties. Vanadium extraction from steel slags is well studied, while extraction from secondary sources such as petcoke is not well understood. Vanadium is one of the 50 critical minerals identified by the United States Department of Interior. Considering the annual production of petcoke, it is a valuable secondary source of vanadium, especially in places with no steel production. This review paper critically examines the compositional differences between steel and petcoke slags and the various extraction methods that apply to vanadium production, particularly from petcoke, considering the environmental issues associated with each technique. Information on the characterization of US petcoke is also included to identify specific extraction methods for vanadium.

**Keywords:** vanadium; petcoke; leaching

## 1. Introduction

The United States and the world have seen exponential growth in the need for critical minerals for various applications, ranging from sustainable energy and national defense to modern electronic and medical applications. Vanadium is a vital mineral in the metallurgical and chemical industries. It is used as an additive in the steel and aerospace industries to improve the properties of iron and aluminum, besides being used as a catalyst for the desulphurization of crude oil. Vanadium is essential in alloying with Titanium to improve its ductility and thermal properties [1]. Thus, it is critical in producing high-performance airplanes such as the SR-71. On average, 1 mg of vanadium can be found in human bodies [2]. Vanadium is also considered vital in the proliferation of renewable technologies via vanadium flow batteries, promising large-scale energy storage for distributed energy resources. Vanadium is known to inhibit cholesterol synthesis, regulate intracellular signal transduction, and regulate the activity of critical enzymes [3]. The US net import reliance (NIR) for vanadium as a percentage of its apparent consumption is 94%, coming in various forms from Russia, Brazil, China, and South Africa. As of 2019, the annual US vanadium consumption is 8400 tons, with over 94% of the vanadium being used for steel alloys, nonferrous alloys, and catalysts [4]. Given the strategic importance of vanadium for the United States and the minimal production within the country, the US government designated vanadium as a critical mineral as of 2022 [5].

Common sources of vanadium include vanadiferous slags/vanadium-rich steel slags, stone coal, uranium–vanadium carnotite ores, titanomagnetite Ores, spent catalysts, scrap metals, black shale, tar sand fly-ash, aluminum slags, flexicoke/petcoke, vanadiferous clays, and other sources such as oil fly ash. An estimated 67% of the vanadium produced worldwide comes from vanadiferous steel slags, while another 22% comes from vanadium-bearing ores such as stone coal. The remaining 11% comes from processing the vanadium-bearing oil supply [6–8]. The source of this vanadium in steel slags is the vanadium present

in the titanomagnetites or scrap steel that is put back into the feed stream [9,10]. This vanadium is separated along with the slag in the Basic Oxygen Furnace (BOF), where it is primarily concentrated. US raw steel production as of 2020 is 72.7 million metric tons compared to 1064.8 million metric tons in China over the same period [4,11]. Steel production in China is expected to stay stagnant at the 2020–2021 levels during 2022 [12,13]. With vanadium demand expected to grow with the worldwide expansion of wind and solar energy to generate electricity, exploring vanadium's secondary sources becomes essential [12]. The potential for The US to have its local source of vanadium-rich steel slag to produce sufficient vanadium for local consumption exists. However, ramping up steel production offers the benefits of a market that consumes the local titanomagnetite ores. Making high-quality steel along with vanadium and titanium by-products is highly desirable. Still, it would require federal mandates and subsidies, a discussion of which is outside the scope of this paper.

Carnotite Ores are uranium–vanadium ores found around the Colorado Plateau in the United States. Vanadium is extracted as a by-product of uranium extraction [14,15]. The vanadium production from the carnotite ores has been intermittent and only operational when the market conditions warrant [16]. This may change in the future with increasing interest in nuclear thermal propulsion systems for deep space exploration and the potential increase in demand for uranium [17].

Stone Coal is a vanadium-bearing carbonaceous shale found in China, which accounts for 87% of the total vanadium reserves [18–23]. Stone Coal accounts for around 22% of the total vanadium production in the world [6].

Petroleum coke, commonly known as petcoke, is a solid by-product generated during the cracking and refining of crude oil known to be rich in vanadium. The US is the largest producer of petcoke globally, producing over 300 million barrels every year between 2005 and 2019 [24]. Assuming that all the petcoke produced in the US is green-petcoke with a density of 0.7 kg/dm$^3$, 1% ash content, and 6% $V_2O_5$ content, over 12,300 short tons of vanadium could potentially be extracted each year, which is almost 1.5 times the annual vanadium consumption of the United States [24–26]. Petcoke is widely available across the world, either within landfills or being used as a cheap replacement for coal, making it an attractive source of vanadium. There are a limited number of studies on vanadium extraction from petcoke in the literature.

Therefore, the main aim of the current work is to critically review petcoke as a secondary source of vanadium and identify potential environmentally benign methods of extracting vanadium from petcoke based on existing methods employed for other sources of vanadium.

## 2. Vanadium Extraction Methods

The range of elemental composition of the major sources of vanadium from the literature is provided in Table 1. Petcoke ash has a composition similar to steel slags, although petcoke ash is formed in gasifiers under reducing conditions, and vanadium-rich steel slags experience oxidizing conditions while forming in a BOF converter [27]. They are both high in their V and Fe content while having a moderate amount of Si when compared to carnotite ores and stone coal, which are low in their V content while being high in their Si content. The steel slags and petcoke also have high Ca content. Limestone is added to the iron blast furnace, where the Ca forms a slag with Si, which floats over the liquid iron [28]. Limestone, on the other hand, is added to gasifier slags to improve their flowability [29]. However, petcoke ashes inherently have a moderate amount of Ca, around 15%. The presence of excess lime, however, can lead to greater acid consumption in both cases, unlike other sources of vanadium [30]. Vanadium in the steel slags is associated in spinel form with Ca or Fe [30–44]. The association of V with Fe is common in the low Ca slags, while the opposite is true for the high calcium slags. Vanadium in petcokes can be associated in spinel form with Fe as carnotite or Ca [45,46]. The similar composition of petcoke and steel slags and the forms in which vanadium is present in them make steel slags a reasonable

analog to petcoke with regards to vanadium extraction. This is especially important due to the limited literature available on vanadium extraction from petcoke, while vanadium extraction from steel slags is commonly practiced and well-explored.

**Table 1.** Range of composition of major vanadium feedstocks.

| | $V_2O_5$ | CaO | T-Fe | $SiO_2$ | $Al_2O_3$ | $TiO_2$ | MgO | $Na_2O$ | $MnO_2$ | $K_2O$ | $SO_3$ | $P_2O_5$ | NiO | $UO_2$ | $Cr_2O_3$ |
|---|---|---|---|---|---|---|---|---|---|---|---|---|---|---|---|
| Vanadium rich steel slag [9,14,30,32,34–39,41–44,47–52] | 1 to 16% | 0.8 to 53.7% | 17 to 48% | 6 to 25.1% | 0 to 10.2% | 1 to 44% | 1 to 15.7% | 0 to 0.2% | 0.4 to 11% | - | - | 0 to 3.2% | - | - | 0 to 8.7% |
| Carnotite Ores [14,53–55] | 0.03 to 2% | 0.4 to 3% | 1 to 20% | 40% to 79% | 2% to 16% | 0 to 1% | 0 to 0.5% | 0 to 1% | 0 to 3% | 0 to 3% | - | - | - | 0.05 to 3% | - |
| Stone Coal [14,19–21] | 1% | 3 to 6% | 4% | 50 to 65% | 7 to 12% | - | 1.5 to 2% | 0.1 to 0.6% | - | 2 to 3% | 1 to 3% | - | - | - | - |
| Petcoke Ash [56–64] | 6 to 57% | 1 to 21% | 4.5 to 28.4% | 13.8 to 35% | 6 to 23% | 0.3 to 5% | 0.6 to 3% | 0 to 5% | - | 0 to 9% | 0.75 to 1.6% | 0.32 to 0.8% | 0 to 12% | - | - |

Vanadium is produced by either thermal techniques as ferrous vanadium or as pure vanadium using hydrometallurgical methods [14,65,66]. While ferrovanadium can easily be used to make ferrous alloys of desired compositions, pure vanadium gives much greater control and flexibility with non-ferrous alloys, catalysts, etc. Thermal methods are primarily used for producing ferrovanadium, while this review focuses on the production of pure vanadium [65]. Similar to an electric arc furnace used to concentrate the vanadium for vanadium recovery from BOF slags, thermal methods are also used to concentrate the vanadium into a feedstock, which can then be extracted through hydrometallurgical methods [67,68]. Thus, thermal techniques are not the focus of the current study. The selection of the appropriate vanadium extraction technique depends on the source material and the desired product, as shown in Figure 1.

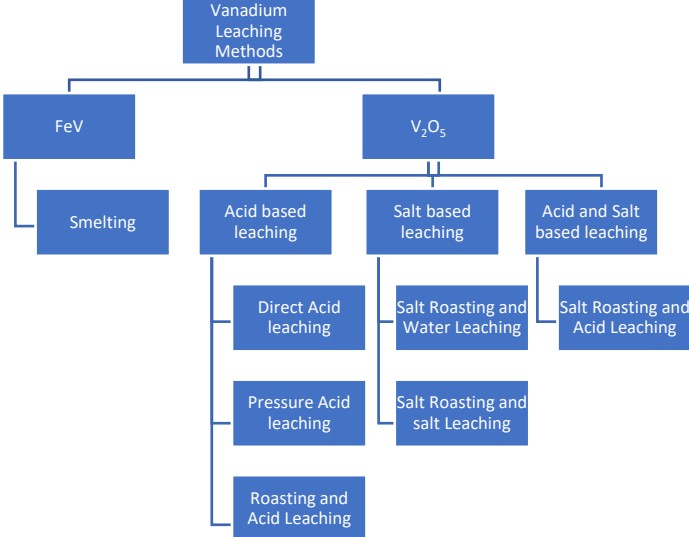

**Figure 1.** Methods for leaching vanadium based on the desired end product.

### 2.1. Physical Beneficiation and Mineral Preconcentration

Physical beneficiation of vanadium for the major sources is typically not required or effective because they are all process solids/residues, which are high in their vanadium content and more amenable for vanadium extraction through hydrometallurgical techniques, while also not being amenable for physical beneficiation due to their chemical makeup [14]. However, with some sources, like titaniferous magnetites, the raw material is first preconcentrated before the extraction of vanadium, either directly or indirectly. Be-

cause vanadium is usually in the magnetite-rich fraction, titaniferous iron ores are ground and then magnetically separated [69]. In the case of titaniferous steel slags, the finer particles (−0.2 mm fraction) tend to be rich in iron/vanadium/titanium fractions when using hydro cyclones, spiral classifiers, or Reichert-cones [69]. For lead–vanadate ores, the high density of lead–vanadates (5.3 to 7.1 g/cm$^3$) vs. the gangue minerals (2.65–2.85 g/cm$^3$) allows for gravity separation for upgrading the ores. Froth flotation is used alongside gravity separation for upgrading the fine-grained ores in mines in Namibia and Zambia for further enhancing the recoveries [70]. This varies depending on the source of vanadium. In the case of LD converter steel slags, vanadium is concentrated in the slag due to its lower density as compared to iron oxide when oxygen is blown through the molten metal [71–73].

## 2.2. Carbon Separation

Before beginning with the vanadium extraction process, any carbon needs to be removed. Holloway et al. [57] showed that carbon content over 15% in the feedstock could significantly reduce the amount of extractable vanadium. When directly leaching petcoke, carbon tends to reabsorb the leached vanadium and forms complexes with vanadium that are harder to leach [74]. Carbon also changes the redox potential of the leach solution, reducing vanadium recovery.

Jack et al. and Sitnikova et al. [58,75] showed that a greater amount of vanadium could be leached by ashing the petcoke below 500 °C. This is thought to avoid trapping the vanadium in silica-rich spheroids due to the melting of low-melting silicates [65]. Most vanadium extraction studies are conducted on slags after removing the carbon, either by thermal or mechanical means such as flotation.

Queneau et al. [76] devised a method of wet pressure oxidation where a petcoke–water slurry was heated to over 200 °C in an autoclave, with pure oxygen passing at above 50 psig pressure and leaching simultaneously with $H_2SO_4$, NaOH, and $Na_2CO_3$. Due to the high dissolution of Si, Al, and other impurities by the sulfuric acid and caustic soda, soda ash was more effective in providing a high-purity leach liquor. Using oxygen at a 300 psig pressure and a temperature of 400 °C, 99% of the vanadium was extracted with a solid-to-liquid ratio (S/L) of 1:100 and 96% extraction using an S/L of 1:10 at 250 °C, respectively. It must be noted that to achieve his high degree of vanadium recovery, the weight loss of carbon was over 90%. Thus, the process could match the results from ashing, and then leach at temperatures lower than what is required for ashing. A significant issue with such a process is the high pressure and temperature (much higher than the boiling point of water) required for vanadium extraction. Another notable thing about the petcoke fraction used in this study is that it was very rich in V ($V_2O_5$ = 62 to 74% in the mineral matter as opposed to 1 to 34% in other petcokes, with the only other outlier being Conn 1995), which makes the samples less representative as compared to the regular petcoke compositions. When the project was scaled up, pressure leaching of tar sands fly ash in Canada was unsuccessful and abandoned due to excessive clogging of the ion exchange columns [66,74].

## 2.3. Vanadium Recovery

A vanadium concentrate through physical beneficiation and pre-concentration goes through salt roasting or calcination in an oxidizing or reducing environment. The vanadium present in the roasted/calcined concentrate is then solubilized using either acids or salt solution or, in the case of salt-roasted samples, leached in water. This leachate goes through a solid–liquid separation before being converted into high-grade vanadium pentoxide, as shown in Figure 2. Acid leaching can leach $V^{+3}$ and $V^{+5}$, while salt leaching can only leach $V^{+5}$ form readily. Pourbaix diagrams are very useful in visualizing this phenomenon and finetuning the leaching method. Pourbaix diagrams or Eh-pH diagrams express the thermodynamically stable chemical species in an aqueous electrochemical solution. pH or the activity of hydrogen ions represents the acidity or basicity of a chemical species. Below pH 7, a solution is acidic, while above pH 7, a solution is basic. Eh, on the other hand, is

the activity of the electrons. Eh is represented in mV as it is also the voltage potential of a solution compared to a standard hydrogen electrode, thus representing the redox potential of a solution. Eh > 0 implies an oxidizing environment, while Eh < 0 represents a reducing environment. To achieve the desired results, such as the precipitation of some species, the Eh or pH may be modified to access a stability zone for the species [77]. It can be seen in Figure 3 that, in a basic solution, to make the vanadium water soluble, the aqueous system needs to be highly oxidizing lest V would precipitate as $V_2O_4$ or $V_2O_3$. The same broadly applies in the case of an acidic system. The only caveat here is that between pH 1 and 3, there is a chance for V to precipitate as $V_2O_5$ if the system is highly oxidizing. Therefore, it may be easier to either drop the pH further or raise the pH so that V dissolves back into the solution.

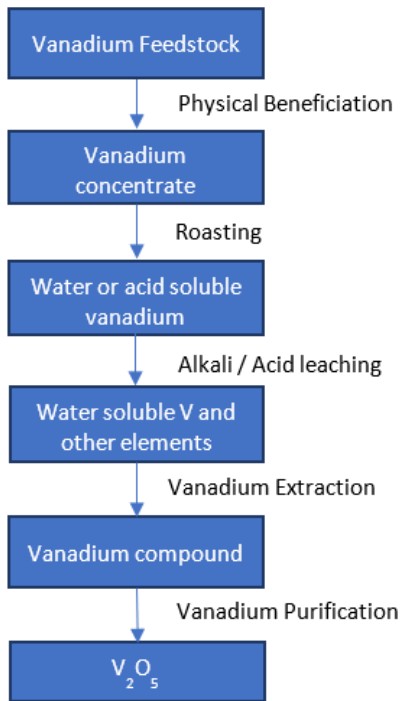

**Figure 2.** General Procedure for extraction of Vanadium from feedstock using a hydrometallurgical method.

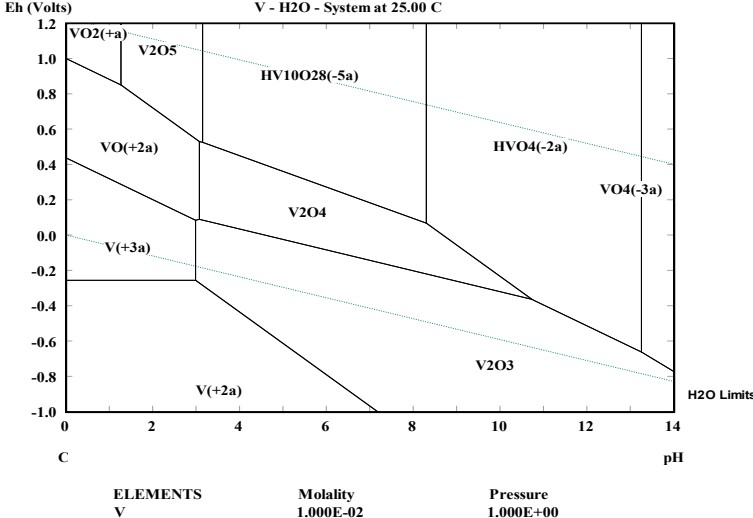

**Figure 3.** Pourbaix diagram for vanadium at 25 °C, 1 bar, and $10^{-2}$ M using HSC Chemistry.

### 2.3.1. Roasting

Roasting is a process with its roots in metallurgy. It refers to heating in the air without fusion; the primary goal is the oxidation of the desired species. Roasting breaks down the spinal structures, thus releasing the vanadium and converting the V into a water, salt, or acid-soluble form [31]. Without roasting, the only way to leach V may be direct acid leaching with concentrated acid or using NaOH solution leaching. In either case, the acid or salt consumption would be very high for low recoveries [14]. While roasting is primarily performed in the presence of sodium salts, calcium salts have also been shown to work if leaching is performed in a sodium salt solution or acid leaching, although this increases the acid consumption significantly, making the process undesirable. Calcium roasting can also lower vanadium recovery, as found by the Russian Tula Factory in the 1970s [39]. In samples with high calcium content, calcination without any salts can break the spinel structures and form calcium vanadates, which are readily acid or salt-solution soluble [31,47].

The sodium salt roasting process typically improves the rate of formation of acid/water-soluble sodium vanadates because $V^{+3}$ in the spinel oxidizes to the $V^{+5}$ form that readily reacts with the sodium salt [57,78]. The roasting process is most effective around the sodium salt's melting point, maximizing the contact between the salt and the vanadium-rich minerals within the ore, thereby aiding the vanadium conversion kinetics [14]. This happens to be around 323 °C for NaOH, around 801°C for NaCl, and 851 °C for $Na_2CO_3$, and 884 °C for $Na_2SO_4$ roasting. The roasting of vanadium is shown in Equations (7) through (10).

$$V_2O_3 + O_2(g) + Na_2CO_3 = 2NaVO_3 + CO_2(g) \qquad (\Delta G_{851\,°C} = -349 \text{ kJ}) \qquad (1)$$

$$V_2O_3 + O_2(g) + 2NaOH = 2NaVO_3 + H_2O(g) \qquad (\Delta G_{323\,°C} = -453 \text{ kJ}) \qquad (2)$$

$$V_2O_3 + O_2(g) + 2NaCl + H_2O = 2NaVO_3 + 2HCl(g) \qquad (\Delta G_{801\,°C} = -225 \text{ kJ}) \qquad (3)$$

$$V_2O_3 + 0.5\,O_2(g) + Na_2SO_4 = 2NaVO_3 + SO_2(g) \qquad (\Delta G_{884\,°C} = -128 \text{ kJ}) \qquad (4)$$

Various studies involving salt roasting for V recovery have also been patented. In a study performed by Gardner [79,80], flexicoke was roasted along with a sodium salt before being leached in water to produce a high V recovery of over 90%. The roasting was found to be effective at 800 °C or higher. Another method applied within the same patent involved ashing the petcoke before directly leaching the mineral matter in an NaOH solution before separating the residue from the leachate, drying, and then fusing with $Na_2CO_3$ before leaching in water. Here, it was stated that the leaching efficiency was sensitive to the ashing temperature, which has to be kept below 600 °C to avoid forming V-Ni refractory, which are difficult to leach V from. In another study by Hass and Hesse [80], $Na_2CO_3$ was mixed with petcoke and gasified in a gasifier. The entrained particles in the gases were separated using a cyclone and then ashed before leaching in 100 °F water. Details on the conditions for ashing were not provided. McCorriston [81] mixed and roasted petcoke and $Na_2SO_4$ from around 700 °C to 900 °C for 2 h to ash the petcoke and fuse the mineral matter; it was then leached in water at 60–100 °C for 3 h. The recoveries of V in this study were maximum at 70%. This may have been because the fused sample was not pulverized before leaching.

However, roasting with salt can also promote other side reactions, which would otherwise not be feasible at low temperatures. This is shown in reactions 5 and 6. These side reactions would consume more salt than theoretically required, and also require dissolving in water and more steps to obtain a higher-grade sodium vanadate solution.

$$SiO_2 + Na_2CO_3 = Na_2SiO_3 + CO_2\ (g) \qquad (\Delta G_{851\,°C} = -67 \text{ kJ}) \qquad (5)$$

$$SiO_2 + Al_2O_3 + Na_2SO = 2NaAlSiO_4 + CO_2\ (g) \qquad (\Delta G_{875\,°C} = -156 \text{ kJ}) \qquad (6)$$

Heat treatment without any salt and leaching in a salt solution could help in avoiding these issues. While there are few studies with regards to heat treatment and salt roasting, Li et al. [36] showed that a 40% solution of ammonium bicarbonate solution was able to reach

over 90% of the vanadium from a steel slag in 140 min of leaching at 70 °C. Li et al. [36] suggested that the vanadium from the trevorite structure, $FeV_2O_4$ was forming $V_2O_5$, which was reacting with Mg and Mn to form magnesium vanadate and manganese vanadate. These vanadium compounds of Mg and Mn reacted with the ammonium bicarbonate to form ammonium metavanadate, which was water-soluble and separated from other impurities, along with the magnesium and manganese salts. What was not evident here was how the vanadium was being separated from the solution.

In feedstocks with high Ca or Mg content, there are two different approaches toward roasting. One is the calcination of the feedstock to convert the vanadium trapped in spinels to calcium vanadates, which are readily acid/salt-solution soluble [31,32,47,49]. This can be seen in Equations (7)–(10). For CaO roasting, 850 °C roasting temperature would lead to optimal results [39,42]. Similar behavior has been observed in slags with moderately high Mg and Mn content [36]. Such leaching methods could recover over 90% of vanadium from the petcoke ash, but it is limited to certain compositions.

$$4FeV_2O_4 + O_2(g) = 2Fe_2O_3 + 4V_2O_3 \qquad (\Delta G_{850\,°C} = -181 \text{ kJ}) \qquad (7)$$

$$V_2O_3 + O_2\,(g) = V_2O_5 \qquad (\Delta G_{850\,°C} = -155 \text{ kJ}) \qquad (8)$$

$$V_2O_5 + CaO = CaV_2O_6 \qquad (\Delta G_{850\,°C} = -122 \text{ kJ}) \qquad (9)$$

$$V_2O_5 + 2CaO = Ca_2V_2O_7 \qquad (\Delta G_{850\,°C} = -262 \text{ kJ}) \qquad (10)$$

The other technique for processing high Ca vanadium feedstock is to roast the feedstock with pyrite. This is covered in more detail in Section 2.3.2, under acid leaching.

Effect of Roasting Temperature

This is primarily due to the increased kinetics of the reaction at higher temperatures. The optimum roasting temperature is defined by various factors, including the kinetics of the formation of water-soluble vanadium and slagging temperature of the petcoke/steel slag. Si and Al content, as well as Ca, define the slagging temperature, which can limit the recovery of vanadium by forming a non-reactive glassy layer over the vanadium. For steel slags with low Ca and Si content, it has been shown that using $Na_2CO_3$, for −200 micron and −45 micron steel slag particles, roasting between 850–1200 °C, $V_2O_5$ recovery increases with temperature marginally up to 94%. However, for −45 micron slag, $Al_2O_3$ and $SiO_2$ recoveries are reduced significantly. However, for −200 micron slag, $Al_2O_3$ and $SiO_2$ recoveries increase gradually with temperature while staying much lower than the −45 micron case [51]. In slags with high Si content, vanadium recovery when salt roasting with $Na_2CO_3$ and water leaching maximizes at a roasting temperature of approximately 700 °C at 90% before dropping steeply around 800 °C due to the sintering of Si [41,43]. A similar drop in V extraction has been observed in high Si and Al tar sands fly ashes above 900 °C, again due to sintering [57]. Holloway et al., 2004 [57] also noted that to achieve a similar vanadium recovery, less salt would be required at higher temperatures as compared to lower temperatures. The sintering in high Al, Si ash also affects vanadium extraction using sulfuric acid [82]. On the other hand, roasting the slag with NaOH and leaching it with water maximizes vanadium recovery at close to 100% at a roasting temperature over 600 °C [37]. When the same high Si, Ca slag is heat treated without any sodium salt, maximum vanadium recovery of around 90% in a $Na_2CO_3$ solution is achieved when roasting at around 850–950 °C before the sintering of the slag drops the vanadium recovery [42,49]. For high Si and Ca slags roasted with Ca and leached in dilute $H_2SO_4$, vanadium recovery is maximized between 80–90% at a roasting temperature of around 800 °C [35,39]. The formation of $CaSO_4$ can limit vanadium recovery in such cases because $CaSO_4$ forms an inert layer with limited solubility in water. In the case of high Si Syrian petcoke, however, roasting with $Na_2CO_3$ leaches around 65% V at around 400 °C, dropping to less than 7% when leaching in a $Na_2CO_3$ solution [64]. This is because of the slagging of the petcoke ash at a much lower temperature than 850 °C, potentially contributed by

the higher Ca and Al content of the petcoke. For slags with high Ca content, vanadium recovery is maximum at around 90% at 680 °C when heat treated for five hours along with pyrite ($FeS_2$) without any sodium salt in excess air while leaching with $H_2SO_4$ [30]. When high Ca slag is roasted with $Na_2CO_3$ and leached in a solution that is a mix of $Na_2CO_3$ and NaOH, V recovery is maximum at around 1000 °C, before going down [52]. Thus, high Si, Al, and Ca petcoke is roasted below 850 °C to limit the slagging of the petcoke to maximize the vanadium recovery. Roasting with or without salt does not seem to affect vanadium recovery if the saltless roasting product is leached in a Na-salt solution. The smaller particle size of the feedstock assists in the greater recovery of vanadium.

Effect of Roasting Time

Typically, a roasting time of around one hour has been observed to be sufficient for maximizing the V recovery, with greater roasting time having diminishing returns [20,42,51,52]. However, there are exceptions to the trend. For example, for steel slags with high Ca and moderately high Si studied by Wilkomirsky et al., 1985 [30] a 5 hr roasting time of around 700 °C was required to recover 90% vanadium. This was reported by Madhavian et al., 2006 [52]. This could be best explained by the fact that Wilkomirsky et al. [30] converted the Ca into $CaSO_4$ during the roasting process by adding pyrite to the mix before acid leaching. The $CaSO_4$ formed an inert layer over the V, greatly reducing the kinetics of V recovery, while Mandhavian et al., 2006 [52] stuck to $Na_2CO_3$ roasting and leaching in a NaOH and $Na_2CO_3$ solution, thereby avoiding the formation of the inert $CaSO_4$ layer.

Holloway et al., 2004 [57] showed that, typically, 2 to 3 h of roasting time between 850 and 900 °C are required for Suncor fly ash. Greater roasting time seemed to indicate no further improvement in the vanadium leachability. Holloway et al., 2004 [57] attributed this behavior to the breaking down of the spinal structures, thus releasing and exposing the vanadium.

While little else has been talked about in the petcoke literature about the effect of roasting time, studies relating to vanadium-bearing steel slags have extensively covered the role of roasting temperature. The leaching efficiency of vanadium can increase with increasing leaching temperature, as shown by Li et al., 2015 [41], Lie, Xie 2012 [42], Madhavian et al., 2006 [52], and Yan et al., 2016 [49]. There is a drop in the leachability of the vanadium past a certain roasting temperature, depending on the feedstock composition. This drop has been attributed by the authors to the low-temperature eutectics that melt, forming a glassy phase that traps the vanadium and stops further leaching. Madhavian et al., 2006 [52] identified this glassy phase as some form of calcium–sodium silicate, which he detected using XRD for a sample roasted at 1000 °C and 1100 °C.

2.3.2. Leaching

Leaching of vanadium can be performed either using water, acids, or alkalis/salts. Salt leaching is usually more selective when using sodium salts that are less caustic than NaOH. Salt-roasted ores are leached using either water, an alkali solution, or acids [83]. Acid leaching tends to be non-selective typically but also generally extracts the maximum amount of vanadium from samples that have lower calcium content. Leaching can also be performed directly on the feedstock without roasting, although this typically requires strong acids or bases and temperatures above the boiling point of water and high pressures. Vanadium recovery can also be lower in such cases.

Salt Leaching

The idea behind salt-based leaching is to convert vanadium into a water-soluble form. This involves oxidizing vanadium to $V^{+5}$, either by roasting along with a sodium salt or roasting directly and then leaching in the sodium salt solution. This is typically achieved by roasting with a sodium salt, such as NaOH, $Na_2CO_3$, NaCl, or $Na_2SO_4$, to convert the vanadium to $NaVO_3$ or $Na_4V_2O_7$ [14], both of which are water-soluble. One of the primary factors in choosing one sodium salt over another is the availability of the salt,

cost, wear, and tear caused by using the salt for leaching, and their selectivity. While NaCl is easily available at a low cost and is still widely used, a major issue with using salt is the production of $Cl_2$ gas or HCl [9,14,32]. These can be highly corrosive on the leaching equipment, besides being harmful to the personnel operating the equipment. Special scrubbers are required to clean the exhaust gases to remove the chlorine. Similar issues occur when using $Na_2SO_4$ [9,14,32]. However, studies have shown that $Na_2SO_4$ is also the most selective among sodium salts for leaching vanadium. In the case of NaOH, it is highly corrosive to the leaching equipment, besides being harmful to plant and animal life. This can lead to higher downtime for equipment repair and replacement, besides requiring special measures for handling. $Na_2CO_3$ is far less caustic than NaOH, making it less corrosive to the leaching equipment.

For CaO-baked samples, the optimal temperature for $Na_2CO_3$ leaching is between 90 and 95 °C, as shown in Equations (11) and (12) [32,49]. $Na_2CO_3$ leaching is highly desirable when the CaO-baked sample also has a high Si content. An undesirable side reaction of CaO baking is that it can significantly reduce vanadium recovery by the formation of silicic acid, which is pronounced in dilute acid systems and forms a gel that is highly acid resistant, as shown in Equation (13) [47].

$$CaV_2O_6 + Na_2CO_3 = 2NaVO_3 + CaCO_3 \qquad (\Delta G_{90\,°C} = -42\ \text{kJ}) \qquad (11)$$

$$Ca_2V_2O_7 + 2Na_2CO_3(a) = 2CaCO_3 + Na_4V2O_7 \qquad (\Delta G_{90\,°C} = -0.5\ \text{kJ}) \qquad (12)$$

$$CaSiO_3 + H_2SO_4 + H_2O = CaSO_4 + H_4SiO_4 \qquad (\Delta G_{90\,°C} = -164\ \text{kJ}) \qquad (13)$$

Besides Na-salt, $NH_3$ has also been used for directly leaching vanadium. In the patent filed by Bhaduri and Zestar [84], ammonia was used to directly leach V from a spent hydro processing catalyst. Petcoke, oils, Ni, and V accumulated on the spent hydro processing catalyst was de-oiled, and then underwent pressure leaching with ammonia and water at 120–250 °C and 100–1200 psig for 1–6 h. The slurry was treated with a flocculant at 50–70 °C for 10 min to 1 h with hot aqueous $(NH_4)_2SO_4$ added to suppress the dissolution of ammonium metavanadate. The dissolved Ni and Mo were removed using S/L separation. Then, more hot water was added to the slurry to leach the vanadium followed by another S/L separation to separate the vanadium-rich leachate. Over 90% V could be recovered. The ammonia use in this technique is high, as is the high pressures required for this method.

Acid Leaching

Acid leaching is the process of recovering elements from a feedstock by solubilizing them in an acid. The desired elements could be pretreated to be converted into a form that is more or less readily acid-soluble to separate them from gangue elements. Generally, due to the harsh nature of the acids used, there is low selectivity during leaching, necessitating the use of expensive reagents to extract the vanadium. In addition, the waste generated during the acid-leaching process needs to be cleaned before releasing because it is highly corrosive.

The acid-leaching process works by leaching various elements from the feedstocks as acid-soluble components. This makes this process ideal for feedstocks with low V content [44]. A major advantage of this technique is high vanadium recovery [85]. The vanadium-rich steel slag is typically roasted first to oxidize the vanadium. This is represented in reactions 14 to 16.

$$V_2O_4 + 2H_2SO_4 = 2VOSO_4(a) + 2H_2O \qquad (\Delta G_{60\,°C} = -177\ \text{kJ}) \qquad (14)$$

$$Fe + H_2SO_4 = FeSO_4(a) + H_2(g) \qquad (\Delta G_{60\,°C} = -127\ \text{kJ}) \qquad (15)$$

$$Al_2O_3 + 3H_2SO_4 = Al_2(SO_4)_3(a) + 3H_2O \qquad (\Delta G_{60\,°C} = -233\ \text{kJ}) \qquad (16)$$

In samples with high calcium and magnesium content, including the ones that are CaO baked, the acid consumption can be particularly high and may even inhibit further reaction [85–88]. This is because of the reaction between the Ca present in the slag with

acid to form $CaSO_4$, which forms a stable layer over the particles that the acid has trouble penetrating. For $H_2SO_4$ leaching of CaO baked samples, 85 °C appeared to provide maximum recovery [47]. This is shown in reactions 17 and 20.

$$MgO + H_2SO_4 = MgSO_4 + H_2O \qquad (\Delta G_{60\,°C} = -169 \text{ kJ}) \qquad (17)$$

$$CaO + H_2SO_4 = CaSO_4 + H_2O \qquad (\Delta G_{60\,°C} = -268 \text{ kJ}) \qquad (18)$$

$$2CaV_2O_6 + 6H_2SO_4 = 4VOSO_4(a) + 2CaSO_4 + 6H_2O + O_2(g) \quad (\Delta G_{85\,°C} = -389 \text{ kJ}) \quad (19)$$

$$2Ca_2V_2O_7 + 8H_2SO_4 = 4VOSO_4(a) + 4CaSO_4 + 8H_2O + O_2(g) \quad (\Delta G_{85\,°C} = -684 \text{ kJ}) \quad (20)$$

To avoid excess acid consumption in the high Ca/Mg/Mn samples, typically, pyrite is added and roasted together with the slag to convert the Ca to $CaSO_4$ and Mg to $MgSO_4$ before leaching in dilute sulfuric acid [14]. The slag is then ground to break up the $CaSO_4$ and expose more of the vanadium to be acid leached [44,85,86]. This is expressed in reactions 21 to 23.

$$2FeS_2 + 5.5O_2(g) = Fe_2O_3 + 4SO_2(g) \qquad (\Delta G_{600\,°C} = -1534 \text{ kJ}) \qquad (21)$$

$$CaO + SO_2(g) + \frac{1}{2}O_2(g) = CaSO_4 \qquad (\Delta G_{600\,°C} = -264 \text{ kJ}) \qquad (22)$$

$$MgO + SO_2(g) + \frac{1}{2}O_2(g) = MgSO_4 \qquad (\Delta G_{600\,°C} = -144 \text{ kJ}) \qquad (23)$$

Vitolo et al. [82] was able to circumvented the issue of non-selective leaching when using acid leaching by oxidatively precipitating $V_2O_5$ directly using $NaClO_3$ as an oxidating agent. As can be observed in Figure 3, $V_2O_5$ can precipitate under highly oxidizing conditions between pH 1 and 3. Overall $V_2O_5$ yield was around 80% with 80% grade at 850 °C ashing temp. A higher grade of $V_2O_5$ could be achieved at lower ashing temperatures; however, the recoveries dropped to around 65%. Ashing temperatures above 850°C dropped both the grade and yields of $V_2O_5$. This may be due to the formation of V-Ni refractories, which dropped the V recovery into the leachate during acid leaching. In addition, at 1150 °C, ash fusion was observed in the SEM images.

Sitnikova et al. [75] showed that adding an oxidant during the leaching process improved the vanadium recovery.

Effect of Leaching Temperature

Increasing the leaching temperature increases the leaching kinetics, thus leading to the greater recovery of vanadium in a water-soluble form with increasing temperature [14,34,42,44,49,52,89,90]. In general, it is possible to extract salt-roasted slag at 95 °C or lower in the water. Heat-treated slag can typically be extracted in sulfuric acid around 70 °C. Higher leaching temperatures are required when pressure acid leaching. Queneau et al., 1989 [76] showed this trend under high temperatures (200–270 °C) and high pressures (300 psig) using salt leaching in an oxidizing environment, although the effect was minimal due to the fast kinetics and limited diffusion effects at the conditions, as described. He was able to recover up to 90% V by pressure salt leaching. Similar tests were performed by Gao et al., 2018 [34] with close to 80% vanadium recovery at temperatures around 200 °C vs. around 50% at 130 °C.

Effect of Leaching Time

Typically, by increasing leaching time, the recovery of vanadium increases [34,38,44,52,78,89,91]. However, past 1–2 h, the recovery is asymptotic, with minimal gains with salt leaching, with longer leaching times for lower leaching temperatures, or when the feedstock was not salt-roasted or was not carbon-free, as can be seen in the study by Navarro et al., 2007 [86] for oil fly ash feedstocks. Leaching times tend to be typically higher when direct pressure leaching vs. when the samples are roasted before leaching. This is probably

because of the slow kinetics of conversion at the leaching temperatures, usually below 100 °C, vs. roasting temperatures, which at typically between 700 and 1000 °C.

Microwave and Ultrasound Assisted Leaching

Microwave and ultrasound assisted leaching methods have been gaining traction in recent years. It has been found that microwaves heat the particles from within, and quickly so, leading to thermal fractures forming within the particles, thus creating more surface area for leaching [92]. Ore minerals are typically good absorbers of microwaves, while gangue minerals such as silicates are poor absorbers of microwaves. On the other hand, ultrasound leads to the formation of cavitation bubbles, which can raise the temperature and pressure in highly localized bubbles to as high as 5000 °C and pressures as high as 1000 atm. When these bubbles collapse, the temperature drop is instantaneous, and the bulk temperature more or less remains the same. The collapse of the bubbles formed near the surface of a particle leads to the formation of jets of close to 400 km/h. These jets impinge on the surface of the particle, breaking it down and exposing new surface areas [93]. Together with microwave-assisted leaching, ultrasound ensures a high rate of reaction and a high vanadium recovery, even when using carbon rich samples. Application of the combination of the two methods have shown encouraging results with a carbon rich petcoke using alkali leaching [94,95]. Similarly high recoveries were found when acid leaching vanadium slags [96].

## 3. Environmental Issues with the Various Vanadium Extraction Methods

There are multiple environmental issues associated with the existing leaching processes used in industry. NaCl leaching leads to the formation of $Cl_2$ and HCl. $Cl_2$ is known to cause severe damage to vegetation at over 1.38 ppm and is an irritant to the mucosal membrane in animals [97]. It is also known to be extremely corrosive alongside hydrogen chloride. $SO_2$ formed when roasting with $Na_2SO_4$ is known to cause acid rain, which is corrosive and leaches the nutrients out of the soil as well as trees, turning the soil barren and killing the trees. It also leaches aluminum into the soil, which can end up in water bodies and be dangerous for plants and animals. It can also raise the pH of water bodies and increase the pH, thereby killing aquatic life [98]. Sulfuric acid is highly corrosive and can cause burns to plants and animals alike. As well as changing the pH of water bodies and leaching soil, it can also cause acid rain and leach heavy metals into water bodies, thus polluting them. NaOH is known to be highly corrosive and can affect the pH of water bodies.

Thus, considering the various environmental issues associated with the extraction of vanadium, it is desirable to limit the use of acids as well as salt roasting while leaching vanadium. Using heat treatment of the ores followed by salt leaching with sodium carbonate could open the opportunity for the extraction of vanadium. A US petcoke was classified to identify potential methods of extracting vanadium from it.

## 4. Characterization of the Petcoke Being Studied

Because the literature on US petcokes is practically non-existent, the authors have included the characterization of the petcoke being studied by them to assist the readers. The petcoke under study was sourced from an entrained flow reactor that used petcoke sourced from the US refineries. The petcoke has experienced temperatures over 1300 °C and was partially gasified and decarburized before being recovered. Table 2 shows the composition of the petcoke ash analyzed using XRF. The petcoke was ashed at 750 °C over 12 h in a muffle furnace open to air before being analyzed in the XRF. The petcoke contains 86% fixed carbon. Calcium oxide was added to the petcoke ash to improve the flowability of the slag to make its removal easier while also binding the sulfur. This can be seen in the XRD spectra in Figure 4 as well as the SEM-EDS analysis in Figure 5, where the calcium and sulfur form anhydrite. The XRD spectra also shows that iron and nickel form a spinel, trevorite ($NiFe_2O_4$), while most of the vanadium is associated with a spinel of coulsonite

(FeV$_2$O$_4$). This behavior can also be observed in the SEM-EDS images of the petcoke ash for Fe, Ni, and V, as shown in Figure 6. Table 1 illustrates the difference in the composition of the petcoke ash under study vs. contemporary sources of vanadium. The petcoke ash in this study is composed of low Al, and Si content in contrast to most petcoke ashes. Its composition appears to lie between that of petcoke with its high vanadium and nickel content, and vanadium-rich slags with their high iron content.

**Table 2.** Composition of the petcoke under study.

| Element | Fe | Ni | Ca | V | S | Si | Zn | Al | Na | Mo | Rest |
|---|---|---|---|---|---|---|---|---|---|---|---|
| Wt% | 28 | 24 | 10 | 10 | 7 | 6 | 4 | 3 | 3 | 3 | 2 |

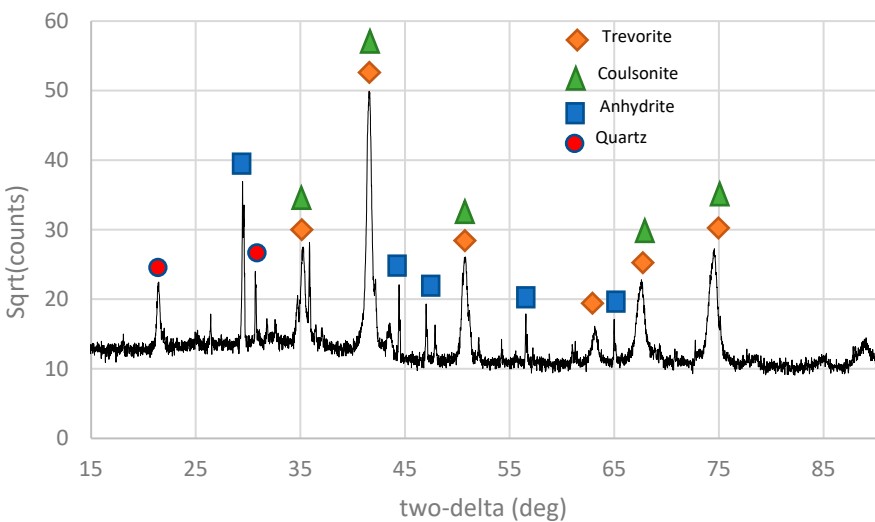

**Figure 4.** XRD Spectra of petcoke ashed at 750 °C.

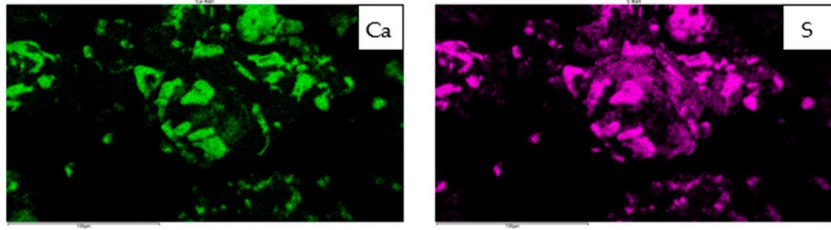

**Figure 5.** SEM-EDS spectra of petcoke ashed at 750 °C, showing Ca and S distribution.

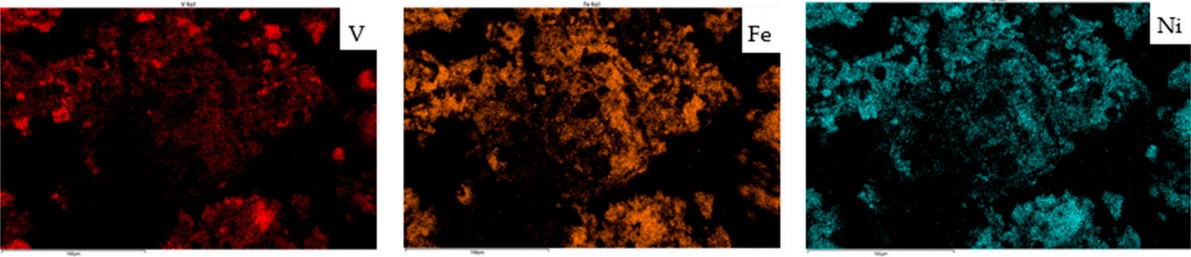

**Figure 6.** SEM-EDS spectra of petcoke ashed at 750 °C for V, Fe, Ni distribution.

The particle size distribution of the petcoke ash performed using a laser particle sizer showed that it was under 100 microns. This implies that the petcoke ash would not require further grinding or pulverization before any salt-roasting or leaching tests.

## 5. Recommendation for Leaching Vanadium from Petcoke Being Studied

For conventional leaching techniques, the petcoke would need to be ashed under 500 °C to remove all the carbon before further processing for V recovery. Given the low Si and Al content of the petcoke ash under study and the moderate Ca content within the petcoke ash, which is primarily associated with sulfur, salt roasting, or direct salt leaching, would present the ideal solution for leaching vanadium from petcoke. Under the roasting conditions, only NaOH would form water-soluble salts with Fe but does not form any compounds with Ni, as shown in Equations (22)–(25). The lack of selectivity when roasting with NaOH would lead to lower-grade vanadium and would require further purification steps before vanadium can be extracted [86]. Thus, it is desirable to avoid using NaOH for salt roasting the petcoke ash and instead perform salt roasting with $Na_2CO_3$. Another advantage of $Na_2CO_3$ would be the avoidance of the formation of $Cl_2$, HCl, and $SO_2$.

$$Fe_2O_3 + 8NaOH = Na_8Fe_2O_7 + 4H_2O \qquad (\Delta G_{1000\,°C} = 346\ kJ) \qquad (24)$$

$$Fe_2O_3 + 4Na_2CO_3 = Na_8Fe_2O_7 + 4CO_2(g) \qquad (\Delta G_{1000\,°C} = 302\ kJ) \qquad (25)$$

$$Fe_2O_3 + Na_2CO_3 = 2NaFeO_2 + CO_2(g) \qquad (\Delta G_{1000\,°C} = 5\ kJ) \qquad (26)$$

$$Fe_2O_3 + 2NaOH = 2NaFeO_2 + H_2O(g) \qquad (\Delta G_{1000\,°C} = -42\ kJ) \qquad (27)$$

Heat treatment in the air without salt opens the possibility of converting $FeV_2O_4$ to a form that will readily react with a less caustic sodium salt solution, such as sodium carbonate, to form sodium vanadate while improving the grade of the vanadium extracted, as represented in Equations (26) and (27). The potential issue with this method could be greater salt consumption.

$$FeV_2O_4 + O_2(g) = Fe_2O_3 + V_2O_5 \qquad (\Delta G_{800\,°C} = -418\ kJ) \qquad (28)$$

$$Na_2CO_3(a) + V_2O_5 = 2NaVO_3\ (a) + CO_2(g) \qquad (\Delta G_{90\,°C} = -29\ kJ) \qquad (29)$$

Acid leaching with $H_2SO_4$ is another potential opportunity for extracting vanadium from petcoke. However, due to the lack of selectivity and issues with handling sulfuric acid, besides the significant environmental problems associated with it, it would not be recommended.

Microwave and Ultrasonic leaching of the petcoke in a $Na_2CO_3$ solution may present yet another interesting avenue to explore due to their effectiveness with recovering V from carbon rich petcoke without ashing.

## 6. Conclusions

Petcoke is a highly underutilized source of vanadium that has the potential to significantly fulfill the global vanadium demand for years to come. Extracting vanadium from this source would also find a use for the vast amount of petcoke that ends up in landfills and pollutes the groundwater aquafers. It would also be easily available in countries that do not have a steel industry that could provide sufficient vanadium-rich slag for vanadium production. This is compounded by the fact that petcoke is a low-cost replacement for coal in many countries, and incorporating vanadium recovery from the petcoke ash from such plants might be a relatively easy process to establish and add value to a resource that would otherwise be an added cost on the books to landfill.

Few studies explore vanadium extraction from petcoke. This study presents the potential methods of extracting vanadium from US petcokes. Salt roasting and leaching or heat treatment followed by salt leaching with $Na_2CO_3$ could provide the ideal solution with minimal environmental or handling issues. Further studies on vanadium extraction from petcoke are required due to their highly variable composition.

**Author Contributions:** Conceptualization, H.J.; methodology, H.J.; software, H.J.; formal analysis, H.J.; investigation, H.J.; resources, S.V.P.; data curation, H.J., S.V.P.; writing—original draft preparation, H.J.; writing—review and editing, H.J., S.V.P.; visualization, H.J.; supervision, S.V.P.; project administration, S.V.P.; funding acquisition, S.V.P. All authors have read and agreed to the published version of the manuscript.

**Funding:** This research received no external funding.

**Data Availability Statement:** There is no data to share here since the document is a review article.

**Acknowledgments:** Nichole Wonderling—Staff Scientist X-ray Scattering Manager*, for assistance with XRD analysis; Julie Anderson—Technical Staff Member SEM, EDS, MP, EBSD*, with assistance with SEM-EDS analysis. * Materials Research Institute, The Pennsylvania State University.

**Conflicts of Interest:** The authors declare no conflict of interest. The funders had no role in the design of the study; in the collection, analyses, or interpretation of data; in the writing of the manuscript; or in the decision to publish the results.

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
