# Peer review of "A Critical Review of Extraction Methods for Vanadium from Petcoke Ash"

_2673-3994, doi:10.3390/fuels4010005_

Round 1

Reviewer 1 Report

The following points may be addressed in the revised version of the manuscript:

  • Literature related to the modeling aspects may be incorporated into this paper and also the latest research studies should be included in the review.
  • The characterization part may be elaborated in detail.
  • It also appears that the author did not give sufficient effort to manuscript preparation as reflected by inappropriate manuscript format, inappropriate grouping, presenting and explaining SEM results, etc. Also, some minor typographical errors need to be corrected.
According to the above observations, it is strongly recommended that the author consult the "author guideline" and the recent issue of this journal for manuscript preparation to improve the manuscript quality of the revised version.  

Author Response

Thank you very much for your constructive suggestions. Every comment has been addressed, and action has been taken. Please see the attached.

Reviewer 2 Report

I have read the article entitled " A critical review of extraction methods for 1 vanadium from Petcoke ash " written by Hari Jammulamadaka and Sarma V. Pisupati, who carry out their professional research work at John and Willie Leone Family Department of Energy and Mineral Engineering, Center for Critical  Minerals, EMS Energy Institute 40. Academic Activities Building The Pennsylvania State University, University Park, PA 16802, USA

The paper is 27 pages long.

The paper is well organised and contains the following sections: 1 Abstract, 2 Introduction, 3 Vanadium Extraction Methods, 4 Environmental Issues with the various vanadium extraction methods, 5 Classification of the petcoke being studied, 6 Recommendation for leaching vanadium from petcoke, 7 Conclusions .

All sections are well developed.

I have done a reading and review of its parts.ç

I will now go on to explain the areas where I think the article could be improved:

Lines 41-43 An estimated 67% of the vanadium produced worldwide comes from vanadiferous steel slags, while another 22% comes from vanadium-bearing ores such as stone coal. The remaining 11% comes from processing the vanadium- bearing oil supply [6-8].”

Zhang et al (2021) claim that “nearly 88% of vanadium in the world is extracted from vanadium-titanium magnetite deposits”

Zhang et al, 2021. Extraction of vanadium from low-vanadium grade magnetite concentrate pellets with sodium salt. 10.1016/j.jmrt.2021.11.039

Another reference indicates that most of the vanadium comes from vanadium-titanium magnetite deposits:

Summerfield, 2019. Australian Resource Reviews: Vanadium 2018. Geoscience Australia, Canberra. 10.11636/9781925848274

The references used by the authors are no older than those used by Zhang et al (2021). The authors should rephrase this paragraph.

Line 93 Table 1: Range of composition of major vanadium feedstocks

The chemical compounds in the table should be homogenised in a single line, as this would make the table clearer and more understandable.

V2O5

CaO

T-Fe

SiO2

Al2O3

TiO2

MgO

MnO2

Na2O

Cr2O3

P2O5

K2O

UO2

SO3

Line 105 3.1 Physical Beneficiation

Rather than talking about physical beneficiation, the authors are talking about mineral concentration processes, where they talk about crushing, grinding, screening, density sorting and processes such as flotation which are rather chemical in that they use properties that do not depend on shape or density, but on the hydrophobicity of the particles. Emulsifiers and depressants used in flotation play on the surface tensional properties of the particle in question to cause it to rise to the surface with the foams formed by the air bubbles or to be depressed and fall to the bottom and exit through the purges.

I think the title of the section is inappropriate and the authors should change it.

Line 196 there are few studies with regards to heat treatment and salt roasting, Li et al. showed that a 40% solution”

this reference is not consistent with the format used in the rest of the paper, it should be Li et al [36]. In the line 198 the same thing happens, it should be corrected.

Line 230 “again due to sintering [57]. Holloway et al. also noted that to get a similar vanadium recovery”

In the line 230 the same thing happens, it should be corrected.

Line 255-257 “Wilkomirsky et al., a 5 hr roasting time of around 700°C was required to recover 90% vanadium [30]. This was reported by Madhavian et al. [52]. This could be best explained by the fact that Wilkomirsky et al. converted the Ca into CaSO4 during the roasting process by adding pyrite to the mix before acid”

In the line 255 and 257 the same thing happens, it should be corrected.

It is inconsistent to do it one way on line 256 (Madhavian et al. [52] ) and the other two in another. It does not show consistency with the process used in the rest of the document.

In the line 259 the same thing happens, it should be corrected.

In the line 263 the same thing happens, it should be corrected.

Line 267-268 “leaching efficiency of vanadium can increase with increasing leaching temperature, as shown by Li et al. 2015, Lie, Xie 2012, Madhavian et al. 2006, and others [41, 42, 52]. There is a drop in the leachability of”

the citation format within the article shows inconsistencies, for example It is inconsistent to do it one way on line 256 (Madhavian et al. [52] ) and in lien 267-268 in another reference system (Li et al. 2015, Lie, Xie 2012, Madhavian et al. 2006). It does not show consistency with the process used in the rest of the document. It should be corrected.

In the line 271 the same thing happens, it should be corrected.

Line 339 “pressure acid leaching. Queneau et al. 1989 showed this trend under high temperatures (200-270) and”

It is inconsistent to do it one way on line 256 (Madhavian et al. [52]) and in lien 339 in another reference system (Queneau et al. 1989). It does not show consistency with the process used in the rest of the document. It should be corrected.

In the line 342, 348 the same thing happens, it should be corrected.

By doing so, it seems that the authors are recycling lines from another article where the citation format is different.

Lines 372-375 its removal easier while also binding the sulfur. This can be seen in the XRD spectra in Error! Reference 373 source not found. Error! Reference source not found.as well as SEM-EDS analysis in Figure 5Error! 374 Reference source not found. where the calcium and sulfur form anhydrite. The XRD spectra also show 375

The references are missing, the authors should correct this mistake.

overall the review article is very successful, critically reviewing the issue of vanadium production from petcoke. However, I find that a number of important references are missing, especially in some of the sections. The authors should make a new version where some of them are included, as it does not make sense to talk about a review and leave out published articles that deal with this specific topic. I say this because the authors talk about fly ash, steel slag and for example they leave out these references that deal directly with petcoke:

Kondrasheva et al, 2019. The influence of leaching parameters on the extraction of vanadium from petroleum coke. 10.1080/10916466.2019.1590406

Zhang et al, 2020. Removal of vanadium from petroleum coke by microwave and ultrasonic-assisted leaching. 10.1016/j.hydromet.2019.105168

Huang et al, 2015. Microwave Leaching of Vanadium from Petroleum Coke. 10.1080/10916466.2015.1043465

Vitolo et al, 2000. Recovery of vanadium from heavy oil and Orimulsion fly ashes. 10.1016/S0304-386X(00)00099-2

Sitnikova et al, 1990. Role of thermal oxidation in the process of vanadium extraction from refinery coke (solid oil residues). 10.1016/0031-6458(90)90017-A

As this is a critical review, the authors should have reviewed existing patents, as they are as close to industrial reality as possible in protecting processes for obtaining vanadium from various resources. For example:

US4816236A Recovery of vanadium and nickel from petroleum residues

Authors are encouraged to do a little research on the topic among the patents to be addressed in the corresponding sections of the article.

Some figures are not shown correctly in the article; this problem should be corrected.

Author Response

Thank you very much for your constructive suggestions. All the comments were addressed, and action was taken as outlined in the attached document.

Reviewer 3 Report

Dear Authors,

In my opinion, the presented review of the vanadium extraction from petcoke ash is very well written and comprehensible. The results are clear and very well summarized. In my opinion, the topic of the paper is to be considered as a high interest if we think about use of vanadium in various steel alloys and the possibility to extract this material with less environment impact and with less energy consumption. The idea of extracting vanadium from petcoke seems to represent great interest if we take into account that petcoke could be a low-cost replacement for coal. After carefully reading the submitted paper I would recommend it for publication in present form.

Author Response

Thank you very much for your time in reviewing the manuscript and providing your valuable opinion.
